# Potential of Natural Products in Hangeshashinto Water Extract on the Direct Suppression of Stomatitis Induced by Intra-/Extracellular Advanced Glycation End-Products

**DOI:** 10.3390/ijms26189118

**Published:** 2025-09-18

**Authors:** Takanobu Takata, Junji Moriya, Katsuhito Miyazawa, Shinya Inoue, Sohsuke Yamada, Jia Han, Qian Yang, Xin Guo, Shuichi Mizuta, Takeshi Nakahashi, Nobuyuki Onai, Hiroyuki Nakano, Togen Masauji, Yoshiharu Motoo

**Affiliations:** 1Division of Molecular and Genetic Biology, Department of Life Science, Medical Research Institute, Kanazawa Medical University, Uchinada 920-0293, Ishikawa, Japan; 2Department of Pharmacy, Kanazawa Medical University Hospital, Uchinada 920-0293, Ishikawa, Japan; masauji@kanazawa-med.ac.jp; 3Department of General Internal Medicine, Kanazawa Medical University, Uchinada 920-0293, Ishikawa, Japan; moriya@kanazawa-med.ac.jp (J.M.); smizuta@kanazawa-med.ac.jp (S.M.); tkn@kanazawa-med.ac.jp (T.N.); 4General Medical Center, Kanazawa Medical University Hospital, Uchinada 920-0293, Ishikawa, Japan; 5Department of Urology, Kanazawa Medical University Hospital, Uchinada 920-0393, Ishikawa, Japan; miyazawa@kanazawa-med.ac.jp (K.M.); s-inoue@inoueiin-kusatsu.or.jp (S.I.); 6Inoue Iin Clinic, Kusatsu 525-0034, Shiga, Japan; 7Department of Pathology and Laboratory Medicine, Kanazawa Medical University, Uchinada 920-0293, Ishikawa, Japan; sohsuke@kanazawa-med.ac.jp (S.Y.); hanj227@kanazawa-med.ac.jp (J.H.); 8Department of Pathology, Kanazawa Medical University Hospital, Uchinada 920-0293, Ishikawa, Japan; 9Department of Spleen and Stomach Diseases, First Affiliated Hospital of Hebei University of Chinese Medicine, Shijiazhuang 050011, China; yang0311qian@126.com; 10Hebei Key Laboratory of Turbidity Toxin Syndrome, Shijiazhuang 050011, China; tianqi11211216@163.com; 11Research Center, First Affiliated Hospital of Hebei University of Chinese Medicine, Shijiazhuang 050011, China; 12Department of Immunology, Kanazawa Medical University, Uchinada 920-0293, Ishikawa, Japan; onai@kanazawa-med.ac.jp; 13Department of Oral and Maxillofacial Surgery, Kanazawa Medical University, Uchinada 920-0293, Ishikawa, Japan; nakano-h@kanazawa-med.ac.jp; 14Department of Internal Medicine, Fukui Saiseikai Hospital, Wadanaka 918-8503, Fukui, Japan

**Keywords:** stomatitis, Hangeshashinto, Kampo medicine, traditional Chinese medicine, water extract, gargle, lifestyle-related disease, advanced glycation end-products (AGEs), dietary AGEs, natural products

## Abstract

Oral mucositis (stomatitis) is a painful condition that affects the mouth lining. Kampo medicines (e.g., Hangeshashinto [Chinese name, Ban-Xia-Xie-Xin-Tang], Orento, and Orengedokuto) have been widely used to treat stomatitis, such as gargling with Hangeshashinto. However, the mechanisms by which Kampo medicines work are not widely understood due to their oral administration and the subsequent digestion, absorption, and metabolization of their components. Stomatitis is associated with advanced glycation end-products (AGEs) in patients with lifestyle diseases, and can be induced by both intra- and extracellular AGEs (blood and dietary AGEs). Various natural products inhibit intracellular AGE generation and suppress cytotoxicity, such as inflammation caused by extracellular AGEs. This review summarizes 19 natural products identified in the Hangeshashinto water extract and 16 natural products identified in the crude drug extract. The data show that several natural products, such as glycyrrhizin, baicalin, 6-shogaol, quercetin, epigallocatechin-3-galate, and genistein, inhibit intracellular AGEs and suppress extracellular AGE inflammation. Furthermore, several natural products in the Hangeshashito water extract can suppress cytotoxicity in stomatitis.

## 1. Introduction

Stomatitis is a painful condition that affects the mouth lining, and there are numerous types and causes, such as traumatic stomatitis [1], aphthous stomatitis [2], and chemotherapy-/radiotherapy-induced stomatitis [3,4]. Aphthous stomatitis is diagnosed in the clinical stage when the cause of stomatitis has not been determined. The proliferation of viruses and bacteria, exposure to cytotoxic materials such as nicotine, and other lifestyle factors can cause stomatitis [5], indicating that it is associated with lifestyle-related diseases (LSRDs). Kampo medicines, which are based on traditional Chinese medicines, such as Hangeshashinto (Chinese name, Ban-Xia-Xie-Xin-Tang) [5,6,7,8], Orento [5,9], Orengedokuto [5,9], Inchinkoto [5,9], Byakkokaninjinto [5], Juzentaihoto [5], Hochuekkiito [5], and Shosaikoto, have been widely used to treat and prevent stomatitis [5]. In Japan, Hangeshashinto is widely used to treat stomatitis, gastrointestinal dysfunction, and nausea [5,6,7,8,9]. Data elucidating the effects and side effects of Kampo medicines have been accumulating for more than two thousand years. However, the mechanisms of Kampo medicines have not yet been fully elucidated due to the high number of natural products contained in each Kampo medicine and the digestion, absorption, and metabolization of natural products by oral administration [10]. While the oral administration of Hangeshashinto to patients with stomatitis, gut dysfunction, and nausea is described in traditional Chinese medicine [11,12,13,14,15], the treatment and prevention strategy for stomatitis described in Kampo is gargling and coating of the mouth with the Hangeshashinto water extract [16,17,18,19]. When gargled or coated on the mouth, natural products in the Hangeshinto water extract may directly protect the oral epithelial cells, thus avoiding digestion, absorption, and metabolism pathways [10]. Oh et al. and Endo et al. analyzed the components in the Hangeshashinto water extract using three-dimensional high-performance liquid chromatography (3D-HPLC) [6,20], and 19 natural products, such as liquiritigenin [21], glycyrrhizin [22], baicalin [23], and 6-shogaol [24,25], were identified. Because these natural products inhibit the production of prostaglandin E2 (PGE2), they can be used as a direct protection of oral epithelial cells [21,22,23,24,25]. In this review, we have focused on advanced glycation end-products (AGEs) associated with LSRDs, including diabetes mellitus (DM), cardiovascular disease (CVD), hypertension, fatty liver, and stomatitis [26,27,28,29]. AGEs originate from saccharides (e.g., glucose and fructose), their metabolites, and non-enzymatic production (e.g., methylglyoxal, glyoxal, and glyceraldehyde) [26,27,28,29]. These compounds react with proteins to produce AGEs; however, methylglyoxal, glyoxal, and glyceraldehyde can also be produced from lipids such as glycerol. Consequently, the definition of AGEs may be modified in the future [26,27,28,29]. Three major free types of AGE, *N*^ε^-carboxymethyl–lysine (CML), *N*^ε^-carboxyethyl–lysine (CEL), and *N*^δ^-(5-hydro-5-methyl-4-imidazolone-2-yl)-ornithine (methylglyoxal-hydro-imidazolone, MG-H1), which are modified proteins, have been reported. These AGE-modified proteins are generated in the human body, and their dysfunction might affect various LSRDs [26,27,28,29]. Intracellular AGEs can be generated and accumulate in oral epithelial cells and esophageal and gastric epithelial cells [30,31,32]. Consequently, intracellular AGEs may induce diseases such as stomatitis. In contrast, extracellular AGEs, which are leaked into body fluids such as blood, urine, and saliva from various cells that produce intracellular and dietary AGEs, can induce cytotoxicity via receptors for AGEs (RAGE) and toll-like receptor 4 (TLR4) [26,27,28,29]. The major dietary AGEs are CML, CEL, and MG-H1 [10,27,28,29]. RAGE and TLR4 are expressed on oral epithelial cells and other cells in various organs [33,34,35]. Extracellular AGEs can be transported and expressed on oral epithelial cells, and the AGEs-RAGE/TLR4 axis induces cytotoxicity such as inflammation [33,34,35]. Extracellular AGEs can cause stomatitis; consequently, this review has focused on natural compounds that can suppress stomatitis. Specifically, the natural products in Hangeshashinto water extract are reviewed, and examples such as liquiritigenin [36], glycyrrhizin [37], baicalin [38,39], and 6-shogol [40,41,42], which inhibit the production of AGEs in vitro and suppress the AGEs-RAGE/TLR4 axis signaling, are discussed. The literature on natural products in seven crude drugs (Hange, Ogon, Oren, Ninjin, Kankyo, Taiso, and Kanzo), which are contained in Hangeshashinto, has also been reviewed. We discuss the natural products in each crude drug, such as quercetin [43], epigallocatechin-3-gallate [44], genistein [45], jasmonic acid [46], and gallic acid [47]. The relationship between each compound that occurs or promotes LSRDs and Kampo medicines remains unclear, although the effects of Kampo medicines have been revealed [10].

## 2. Various Types of Stomatitis

Stomatitis can be caused by various factors (Figure 1). Traumatic stomatitis can be caused by a traumatic physical stimulation, such as a burn [1,48], which causes oral epithelial cells to peel off. In contrast, aphthous stomatitis is characterized by ulcers, insomnia, psychological stress, and other factors, such as the excessive intake of saccharides and lipids, and we predict that AGEs specifically can induce stomatitis (See Section 6). Although the mechanisms of aphthous stomatitis remain unclear [2,49], diagnosis has been performed when an obvious factor in the patient’s mouth or other organs is not observed. Viruses and bacteria can cause herpetic and candidal stomatitis [50,51,52,53]; however, inflammation is usually induced to inhibit their proliferation. Protection against virus and bacteria proliferation is associated with lifestyle, and related stomatitis may be due to LSRDs. Nicotine is one of the main chemicals in all forms of tobacco [54,55]. Nicotine stomatitis, a LSRD, is characterized by an oral lesion, usually located on the hard palate, which develops due to the heat and chemical irritation caused by tobacco products. Vitamin B_1_ deficiency inhibits the recovery of oral epithelial cells against damage and induces stomatitis [56]. Furthermore, vitamins B_2_, B_6_, and B_12_ are required to produce proteins in oral epithelial cells and maintain metabolism, and thus, deficiencies in these vitamins may also induce stomatitis [56,57]. We believe that cases of stomatitis induced by B_1_, B_2_, B_6_, and B_12_ deficiencies should be classified as LSRDs. Chemotherapy and radiation treatments for cancer suppress the oral epithelial cell cycle, and consequently, damaged cells can induce inflammation leading to stomatitis [3,4]. Furthermore, suppression of the immune system in oral epithelial cells can also promote the proliferation of viruses and bacteria, which can also lead to stomatitis. Because chemotherapy and radiation do not occur in daily life, these cases are not classified as LSRDs [3,4,8,15].

## 3. Treatment of Stomatitis with Hangeshashinto and Predicted Mechanisms

### 3.1. Traditional Chinese and Kampo Medicines

To describe the Hangeshashinto treatment for stomatitis, we first introduce the relationship between traditional Chinese medical science and Kampo, which is traditional Japanese medicine [58,59,60,61,62]. Traditional Chinese medical science, which includes health preservation (healthy diet and exercise) [63], diagnosis [62,64,65], moxibustion [66], cupping [67,68], acupuncture [69,70,71], and crude drugs (natural medicines) [6,64,65,72,73,74], has been used for approximately 3000 years. The practices migrated to Japan during the ancient and middle periods, directly from the Chinese mainland via the Korean peninsula. They were adapted to the Japanese population, land, and medical plants, and renamed Kampo [59,60,61,62]. “Kam” or “Kan” in Japanese means “Han” in Chinese, referring to the ancient Chinese empire (from the third-century BC to the third-century AD), and “po” refers to the method of diagnosis and treatment (Figure 2a). Various traditional Chinese medicines involving plants, animal tissues, and minerals are used in Kampo [59,60,61,62,75,76]. Ban-Xia-Xie-Xin-Tang, a set of seven crude drugs, was one of the treatments adopted by Kampo, and referred to as Hangeshashinto (Table 1) [5,6,7,8,11]. In traditional Chinese medical science, Ban-Xia-Xie-Xin-Tang is orally administered, used as a gargle, and as a coating on the oral squamous [11]. The gargle and coating onto the oral squamous cells of Hangeshathinto are described in the *Japanese Pharmacopoeia*, which includes registered Kampo medicines (Figure 2b) [5,6,7,8]. This review has focused on the use of Hangeshashinto as a gargle and coating on oral squamous cells (see Section 3.2).

### 3.2. Hangeshashinto (Ban-Xia-Xie-Xin-Tang) Treatment Effects and Mechanisms

Many Kampo medicines are orally administered water extracts from crude drugs [73,74]. This includes the Hangeshashinto water extract, which is a dry powder (dry powder quality is regulated in the *Japanese Pharmacopoeia*) used to treat oral squamous cells, gut dysfunction, dyspepsia, vomiting, and nausea [5,6,7,8,11]. Hangeshashinto water extract has anti-oxidation, anti-inflammation, anti-bacteria, tissue recovery, pain relief, and intestinal bacteria induction effects [5,6,7,8,11,16,77]. However, the detailed mechanism has not yet been revealed because its various natural products are digested, absorbed, and metabolized in the body’s organs [10]. We predicted that the extract components would directly induce anti-oxidation, anti-inflammation, anti-bacterial, and pain relief when used as a gargle and coating for oral epithelial cells; however, this will require further investigation to elucidate fully [6,7,10,16,77]. Oh et al. reported that Hangeshashinto suppressed the expression of interleukin (IL)-6 and 8 in CAL27 (a human oral epithelial cell line) treated with *Porphyromonas gingivalis* pathogen-associated molecular pattern (PAMP) [6]. Hato et al. revealed that Hangeshashinto suppressed the expression of IL-1α and human β-defensin 1 in normal human oral keratinocytes, which were stimulated with lipopolysaccharide [7]. Hangeshasnito inhibited cyclooxygenase (COX)-2 and suppressed the production of prostaglandin E_2_ (PGE_2_) [16]. A scratch and cell migration test using normal human oral keratinocytes by Uezono et al. revealed that Hangeshashinto induced the secretion of the C-X-C motif chemokine ligand (CXCL) 12 from cells, and secreted CXCL 12 combined with C-X-C chemokine receptor 4 (CXCR4) to promote migration [77]. This effect was mediated by the phosphorylation of extracellular signal-regulated kinase (ERK), and the activity of ERK was associated with increased CXCL12 expression.

### 3.3. Natural Products in Hangeshashinto and Their Anti-Inflammation Effects

Oh et al. and Endo et al. analyzed the components in Hangeshashinto water extract using 3D-HPLC and identified approximately twenty natural products [6,20]. They classified flavonoid, chalcone, alkaloid, and gingerol groups. The flavonoids included liquiritin [6], liquiritin apioside [6], wogonin [6,20], wogonin-7-O-glucuronide [6], baicalin [20], orxylin A [6], and orxylin-7-*O*-glucuronide [6]. The chalcones included isoquritin [6], isoliquiritin apioside [6], and isoquitigenin [6]. The alkaloids included palmatine [6], berberine [6], epiberberine [6], jateprrhizine [6], coptisine [6], and magnoflorine [6]. The triterpene group included glycyrrhizin (glycyrric acid) [6,22]. The gingerol group (one of the monophenolic acid groups) included 6-gingerol [20] and 6-shougaol [6]. The structures of these natural products are presented in Figure 3, Figure 4, Figure 5 and Figure 6. Because the Hangeshashinto water extract inhibited COX-2 and suppressed the production of PGE_2_ [16], we surveyed reports on natural products that show inhibition of COX-2 and suppression of PGE_2_ production. Liquirtin [78], liquiritin apioside [79], liquiritigenin [21], worgonin [16,80], baicalin (baicalein-7-*O*-glucuronide) [16,23,80], orxylin A [81], isoquritin [82], isoquitigenin [82], palmatine [83], berberine [84], epiberberine [85], coptisine [86], glycyrrhizin [22], magnoflorine [87], 6-gingerol [16,24,88], and 6-shougaol [16,24,25] can inhibit COX-2 and/or suppress the production of PGE_2_. Because the Hangeshashinto water extract contains a high concentration of natural compounds, it cannot be suggested with certainty that these identified compounds are the sole cause of the inflammatory effects [6,20]. However, these results may help elucidate the mechanisms of Hangeshashinto.

## 4. AGEs

### 4.1. AGE Origins

Saccharides, saccharide metabolites, and their non-enzymatic reaction products react with proteins to generate AGEs via the Maillard reaction [26,27,28,29,89,90,91,92]. We introduce glucose [26,27,28,29,93,94], fructose [26,27,28,29,93,94], melibiose [95], and ribose [96,97] as major saccharides (Figure 7), and glyceraldehyde, glycolaldehyde, methylglyoxal, glyoxal, and 3-deoxyglucosone as metabolites and non-enzymatic reaction products of saccharides (Figure 8) [26,27,28,29]. Because fructose can be generated from glucose via the polyol pathway and methylglyoxal can be produced from glyoxal, the production routes of AGEs are intertwined [26,27,28,29]. However, we should consider the definition of AGEs because some AGEs may be advanced lipoxidation end-products (ALEs) [29]. AGEs can also be produced from lipids, such as methylglyoxal, which can be made from glycerin [98,99].

### 4.2. Free AGEs

While AGEs can react with proteins to generate AGEs via the Maillard reaction, they can also react with amino acids to produce free-type AGEs (Figure 9 and Figure 10) [26,27,28,29]. We introduced AGEs that were detected and identified in human and/or animal tissues/cultured cells, and body fluids, such as blood, saliva, and urine (Figure 9) [26,27,28,29,100,101,102,103], and AGEs synthesized in a tube [104], whose structure had been hypothesized [105] (Figure 10). Although the AGEs in Figure 9 are the major free types, CML, CEL, MG-H1, methylglyoxal-derived imidazolium cross-link (methylglyoxal–lysine dimer, MOLD), and pentosidine in the blood and urine have been selected as biomarkers for LSRDs or a dairy diet (see Section 4.5.5). Shigeta et al. reported that pyrrolopyridinium–lysine dimer-derived glyceraldehyde 1 and 2 (PPG1 and PPG2) were synthesized from glyceraldehyde and *N*^α^-acetyl-lysine in the tube [104]. However, there are no reports that PPG1 and PPG2 were detected in vivo. Takeuchi et al. hypothesized the structure of two glyceraldehyde-derived AGEs, which they named “Toxic AGEs (TAGE)” in 2004; however, the structure has not been proved [105].

### 4.3. Previous AGE Categories and Potential Improvements

In the middle of the 20th century, a classical category for AGEs was established [26,27,28,29]. Glucose-, glyceraldehyde-, glycolaldehyde-, methylglyoxal-, glyoxal-, and 3-deoxyglusone-derived AGEs were named based on their original compounds as follows: AGE-1, -2, -3, -4, -5, and -6 [26,27,28,29]. However, Litwinowicz et al. detected and quantified melibiose-derived AGEs (MAGE) and determined that MAGE should be categorized as AGE-10 [95]. In contrast, one AGE can be generated from various origin compounds, such as CML, which is generated and produced from glucose, ribose, glycolaldehyde, and glyoxal (Figure 11) [28,96,97]. In the future, AGE categories based on the original compounds may be improved. Some researchers have suggested a unique category based on cytotoxicity and/or LSRDs [105,106,107,108]. Takeuchi et al. named a GA-AGE which was recognized by a polyclonal antibody they prepared, as “TAGE” in 2004; however, the structure emains unclear [105]. Shmkova et al. focused on the pridinium moiety of both glyceraldehyde- and glycolaldehyde-derived AGEs and named them as TAGE [106]. Shen et al. suggested that glyceraldehyde-, glycolaldehyde-, methylglyoxal-, and 3-deoxyglucosone-derived AGEs should be categorized as TAGE because they could directly induce cell damage [107]. Furthermore, Lee et al. used a cytotoxicity analysis to determine that MOLD is a typical TAGE [108].

### 4.4. Structure of AGE-Modified Proteins

AGEs can exist as AGE-modified proteins or animals in the organs [109,110,111,112] and body fluids of humans and animals [113,114,115]. Various methods have been used to identify the structures of AGE-modified proteins, including electrospray ionization–mass spectrometry (ESI-MS) and matrix-assisted laser desorption/ionization MS (MALDI-MS) analysis (see Section 4.6) [109,110,111,112,116,117]. Some AGEs, including free types, can be modified into one molecular protein, as shown in Figure 12a [27,28,29]. Because free-type AGEs generally have modified lysine or arginine residues in amino acid sequences, the location is limited compared with the number of whole amino acids. Although ESI-/MALDI-MS can be used to elucidate this structure, the operation is complex because the peptide that contains more than two free-type AGEs must be detected and identified [27,28,29]. In contrast, free-type AGEs, which include more than two lysine and/or arginine residues, show the potential of the free-type AGE structure in terms of intra-/intermolecular covalent bonding for proteins (Figure 12b) [27,28,29]. In this structure, we believe ESI-/MALDI-MS analysis should be used for identification; however, we understand the limitations of MS technology. The protein database of the ESI-/MALDI-MS equipment cannot be used to detect the structures of peptides that originate from complex proteins combined with free-type AGEs [27,28,29]. In contrast, researchers cannot prove this structure with only anti-AGE antibody analysis. If an anti-AGE antibody recognizes a protein epitope, it does not confirm the difficulty of the inter- or intramolecular covalent bonding capacity [27,28,29,118] (Figure 12b).

### 4.5. Intra-/Extracellular AGEs and LSRDs

#### 4.5.1. Intracellular AGEs and LSRDs

AGEs are generated from saccharides, their metabolites, and non-enzymatic production in the cultured cells [116,119]. Sanavirathna et al. analyzed intracellular AGEs in a pancreatic ductal cell line (PANC-1) treated with glyceraldehyde, and identified argpirimidine-, MG-H1-, and GLAP-modified proteins [116]. The functions of these proteins may be affected by AGE modification. Takahashi et al. reported that intracellular CML-modified proteins increased in a human proximal tubular cell line (HK-2), which was incubated in a high glucose medium, had an Atg5 knockdown, and lysosome function was suppressed by intracellular glucose-derived AGEs [119]. Although their results were in vitro, they indicate a relationship between intracellular AGEs and LSRDs. In contrast, various AGEs (e.g., MG-H1, G-H1-modified proteins) were detected in cardiac tissues of patients (age >75 years) or patients with diabetes mellitus [109,110,111]. These AGEs may suppress cardiomyocyte function to induce CVD. The skeletal muscle in the obese mouse that ingests high-fat and high-saccharide foods accumulated CML- and CEL-modified proteins, and they were associated with adipose degeneration of skeletal muscle [120]. Furthermore, the accumulation of CML and CEL may induce sarcopenia.

#### 4.5.2. AGEs in the Extracellular Matrix and LSRDs

Pentosidine-modified collagens are AGE-modified extracellular matrix proteins [29,121,122]. Intracellular collagens may be modified with free-type AGEs, such as pentosidine, and then secreted and released. Extracellular AGE modification may also occur. Pentosidine-modified collagen showed the dysfunction and may induce [29,121,122].

#### 4.5.3. AGEs in the Blood, Urine, Saliva, and LSRDs

AGEs, including free types, and AGE-modified proteins have been detected in blood [29,123,124,125], urine [29,126], and saliva [29,101]. Many researchers believe that AGEs in the body fluid leak into the blood and lymph vessels, and renal tubules from cells where intracellular AGEs were generated [29]. However, we suggest that AGEs may be generated in the blood/lymph vessel and renal tubule due to the detection of glyceraldehyde [127,128], glycolaldehyde [129], methylglyoxal [130], and glyoxal [130] in the blood. Because these compounds rapidly react with amino acids or proteins, some AGEs may be generated in the blood/lymph vessels and renal tubules. Furthermore, AGEs in foods and beverages have been reported as dietary AGEs (see Section 4.5.4). Various organs, such as the liver, heart, lung, gut, and oral squamous cells, express RAGE and TLR4; AGEs-RAGE/TLR4 signaling can induce dysfunction and cytotoxicity, such as excess inflammation [33,34,35,131,132,133,134]. In contrast, Wang et al. reported that glyceraldehyde-derived AGEs-modified bovine serum albumin, which was the model of AGEs and might contain various AGEs in the blood, induced dysfunction of cardiomyocytes via the suppression of ryanodine receptor 2 activity [135]. Lee et al. revealed that glyoxal-derived imidazolium cross-link (glyoxal–lysine dimer, GOLD) and MOLD induced oxidative damage and inflammation by interacting with RAGE [108,136].

#### 4.5.4. Dietary AGEs and LSRDs

Various types of saccharides are contained in foods and beverages, and they are processed using heat treatments in factories and homes [28,89,133,137,138]. While multiple types of AGEs (free-type AGEs and AGE-modified proteins) can be produced, we acknowledge that CML, CEL, and MG-H1 are widely produced in many foods and beverages [89,133]. Many researchers believe that the AGEs detected in the body fluid will include dietary AGEs, and the AGEs-RAGE/TLR4 axis induces inflammation in various organs, promoting LSRDs such as cancer and gut ulcer [133,137].

#### 4.5.5. AGEs in the Body Fluid as a Biomarker for LSRDs and Dietary Lifestyle

To diagnose LSRDS, such as non-alcoholic heptosteatosis (NASH), hyperuricemia, high-density lipoprotein cholesterol (HDL-C), low-density lipoprotein cholesterol (LDL-C), uric acid, creatinine, IL-1β, IL-6, IL-10, and urinary pH, have been measured in the blood and urine [139,140]. Because AGEs in the blood, urine, and saliva can be detected in humans and animals, various researchers have suggested their use as LSRD [29,95,114,115,140,141] and dietary lifestyle biomarkers [137,138]. Litwinowicz et al. suggested that MAGE in the blood could be a beneficial biomarker for alcoholic hepatitis [95] and insisted that it should be categorized as AGE-10 [95]. CML and MG-H1 are reduced in the blood after kidney transplantation and can thus be used as biomarkers for kidney function [114]. Kato et al. reported that MG-H1, but not CML, CEL, and *N*^ε^-carboxymethyl–lysine (CMA), was elevated in the blood of patients with nephropathy [141]. CML and CEL in the blood increased in obesity model mice fed high-fat and high-sugar diets [114], and various types of AGEs (each structure was not identified) in the blood of NASH model mice were also increased [142]. In contrast, CML, CEL, and MG-H1 in the blood reflect the intake of dietary AGEs and can be used as biomarkers for dietary lifestyle [137,138].

### 4.6. Identification and Quantification of AGEs

We introduce fluorescence, immunostaining, slot blotting, Western blotting, ELISA, GC-MS, ESI-/MALDI-MS, and nuclear magnetic resonance (NMR) as methods to identify and quantify AGEs [26,27,28,29]. AGEs are generally excited at a wavelength of approximately 370 nm, and fluorescence is emitted at approximately 440 nm [143,144]. Although fluorescence cannot be used to differentiate between different AGE structures, it can be used to quantify AGEs in the human skin and blood under non-invasive conditions, and is a beneficial analysis for clinical operations [144]. To analyze basic AGE research, HPLC and fluorescence can be combined, as well as hydrophilic interaction liquid chromatography (HILIC) [145]. Immunostaining, slot blotting, Western blotting, and ELISA analysis generally require anti-AGE antibodies [26,27,28,29]. Although immunostaining, slot blotting, and ELISA can be used to quantify AGEs, they cannot be used to analyze their molecular structures [26,27,28,29]. In the quantification with slot blot analysis, we believed that Takata’s method is beneficial because their lysis buffer (or modified Takata’s lysis buffer) promotes suitable probing of AGE-modified proteins onto polyvinylidene difluoride (PVDF) membranes [26,146,147,148,149]. This method was selected in thirteen studies from 2017 to 2025. Some researchers analyzed fluorescence to quantify AGEs using immunostaining and the ELISA method without using anti-AGE antibodies [117,150]. Western blotting analysis of AGEs can detect various AGE proteins in cell lysates, tissue lysates, and body fluids on the PVDF membrane [120,149]. To quantify AGEs with slot blotting and ELISA, AGE-modified proteins are required, not general recombinant proteins [26,146,147,148,151]. GC-MS [26,152,153,154,155], ESI-MS [26,89,111,123,124,141,156,157], and MALDI-MS [26,89,111,123,124,141,156,157] were used to analyze the mass of free-type AGEs and AGE-modified peptides (Figure 10 and Figure 12). Because free-type AGEs (e.g., CML and CEL) are non-volatile compounds, they should be esterified [26,152,153,154,155]. We believe that ESI- and MALDI-MS analyses are the most suitable for proving that free-type AGEs are modified on one molecular protein and that AGEs with two amino acid residue (e.g., pentosidine, MOLD, GOLD) link proteins with intra- and/or intermolecular covalent bonds, because methods that use fluorescence and anti-AGE antibodies are ineffective (Figure 10 and Figure 12) [26,109,110,111,118]. NMR is used for AGE identification, not quantification [26,28].

## 5. Mechanisms of Inhibition of Intra-/Extracellular AGE-Induced Cytotoxicity by Anti-AGE Compounds

Numerous strategies have been reported for the inhibition of intra-/extracellular AGE-induced cytotoxicity. The carbonyl trap system and activation of glyoxalase-1 can inhibit the generation of intracellular AGEs (Figure 13) [27,28,29]. The former can directly trap the origins of AGEs, which include ketone and aldehyde groups (e.g., glucose, fructose, glyceraldehyde, glycolaldehyde, methylglyoxal, and glyoxal), and reduce the effects of methylglyoxal and glyoxal because they are metabolized by glyoxalase-1 (Figure 7, Figure 8 and Figure 13). Researchers have reported that various compounds show a carbonyl trap system and activation of glyoxalase-1 [27,28,29]. Another method that has been investigated involves inhibiting glucose transport into cells. However, glucose and its metabolites/non-enzymatic products can be generated in cells via the glycolysis and peroxidation of lipids if glucose is not transported. Although AGEs can be degraded by autophagy and the ubiquitin–proteasome system [27,28,29], our understanding of the natural compounds that promote these mechanisms remains unclear. In contrast, anti-AGE compounds may block AGEs-RAGE/TLR4 because they combine with RAGE/TLR4 as antagonists and inhibit the cell signaling pathway (Figure 13) [27,28,29].

## 6. Potential of AGE-Induced Cytotoxicity and Dysfunction for Oral Squamous Cells

### 6.1. Potential of Intracellular AGE-Induced Cytotoxicity and Dysfunction for Oral Epithelial Cells

Oral epithelial cells are located on the oral squamous and typically undergo rapid turnover within 7–14 days [158]. Although intracellular AGEs have not been identified or quantified in oral epithelial cells, existing information about esophageal and gastric epithelial cells indicates that they can be generated and accumulate in oral epithelial cells [10,30,31,32,159]. Esophageal cells are remarkably similar to oral epithelial cells, and Yokoyama et al. reported that the condition of oral epithelial cells may reflect the risk of esophageal epithelial cells [159]. In contrast, intracellular AGEs such as CML and CEL in the esophageal epithelial cells increased in Goto–Kakizaki rats (DM model rats) [31]. The turnover period for gastric epithelial cells is similar to that of oral epithelial cells (7–14 days) [10]. Wang et al. revealed that intracellular AGEs were generated and accumulated to induce damage in gastric epithelial cancer cells and promote tumor formation [32]. Oya-Ito et al. reported that MGO-AGE-modified HSP27 was increased in RGK-1 cells (human gastric epithelial cell line) incubated with high glucose medium, and this MGO-AGE modification may be associated with the phosphorylation of HSP27 [30]. Because phosphorylation regulates the activity of HSP27, which suppresses apoptotic proteins and works as a chaperone system, MGO-AGE modifications may affect these functions [30]. Because glucose and fructose can be transported into oral epithelial cells as well as esophageal and gastric epithelial cells, we believe that intracellular AGEs can be generated and accumulate in oral epithelial cells. The phenomenon of intracellular AGEs in the oral epithelial cells should be further investigated to identify novel oral squamous cell types.

### 6.2. Potential of Extracellular AGE-Induced Cytotoxicity and Dysfunction for Oral Epithelial Cells

RAGE and TLR4 are expressed on oral epithelial cells as well as other organs [10,33,34,35,132]. Because AGEs in the body fluid (e.g., blood and saliva) can directly combine with RAGE and TLR4, AGEs-RAGE/TLR4 signaling can be induced. To date, there are no reports analyzing extracellular AGE-RAGE/TLR4 signaling in vitro or in vivo in oral epithelial cells. Many researchers have tried to elucidate how inflammatory proteins such as high-mobility group box 1 (HMGB-1) and lipopolysaccharides (LPS) induce inflammation and cell dysfunction via the RAGE/TLR4 to investigate inflammation in oral squamous [10,33,34,35,132]. However, inflammation stimulated by the AGEs-RAGE/TLL4 axis can be induced on oral epithelial cells as well as other cells in the liver, lung, gut, and small/large intestine. Oral epithelial cells are more directly exposed to AGEs in saliva and dietary AGEs than the cells in other organs. Oral squamous cells that induce extracellular AGEs-RAGE/TLR4 should be further researched, as well as the involved inflammatory proteins and lipopolysaccharides.

## 7. Natural Products in Hangeshashinto Water Extract Inhibit the Generation of Intracellular AGEs in Stomatitis

Some researchers have suggested inhibiting the generation of intracellular AGEs [28,121,149,160,161,162] and their degradation via the autophagy and ubiquitin–proteasome system [163,164]. To inhibit the generation of intracellular AGEs, the carbonyl trap system can be used against α-carbonyl compounds, the precursors of AGEs, such as glucose, fructose, glyceraldehyde, methylglyoxal, glyoxal, and 3-deoxyglucosone (Figure 7, Figure 8 and Figure 13). Furthermore, glyoxalase-1, which can metabolize methylglyoxal and glyoxal (Figure 8 and Figure 13), can suppress the transportation of glucose into cells [28,121,160,161,162,165,166]. Although the structures of the natural products that show carbonyl trap systems remain unclear, flavonoid skeletal compounds (Figure 3) [28,160,161], chalcone skeletal compounds (Figure 4) [28,160,161], alkaloid (Figure 5) [28,160], resveratrol compounds [28,167], and *p*-coumaric acid, which is similar to 6-gengerol and 6-shogaol in Figure 6, exhibit carbonyl trap effects [28,168]. Froldi et al. found that each glucose, glyoxal, and ribose was incubated with bovine serum albumin (BSA) in the tube. They also investigated the inhibition effects of baicalin for each glucose-, glyoxal-, and ribose-derived BSA [169]. Baicalin can inhibit the production of each AGE-modified BSA. Although Froldi et al. did not suggest whether these effects were induced by carbonyl trapping, we consider it because baicalin is a flavonoid skeletal compound. Furthermore, baicalin shows anti-α-glucosidase activity, which inhibits the production of glucose from polysaccharides and glycoproteins, and reduces the potential that glucose and its intermediates react with amino acid (e.g., lysine, arginine) residues in proteins [169]. Wang et al. incubated glucose and beef enzymatic treatment solution in a tube and quantified CML and CEL [36]. In this study, liquiritigenin, liquiritin, and glycyrrhizin inhibited the generation of both CML and CEL. We predicted that they would show the carbonyl trapping. In contrast, they were able to reduce the production of both methylglyoxal and glyoxal, and these results suggest that they activated glyoxalase-1 [36]. Carnovali et al. reported that the blood glucose level was reduced with the liquiritigenin treatment [170]. Liquiritigenin can inhibit the generation of intracellular AGEs in three ways [36,170]. Alvi et al. reported that glycyrrhizin blocked ribose-derived BSA generation in the tube where ribose and BSA were incubated [37]. Glycyrrhizin belongs to the triterpene structure compound, which contains one carboxylic acid and is glycosylated. The structure may directly inhibit the production of AGEs, though the mechanism remains unclear. Ahmad et al. reported that berberine inhibited the methylglyoxal reaction with human serum albumin (HAS) in a tube [171]. However, glycyrrhizin suppressed glyoxalase-1 in *Plasmodium falciparum*, but its activation up-regulated expression in rats [172,173,174]. Because glycyrrihizin was orally administered and may have been digested, absorbed, and metabolized in previous investigations [173,174], the effects of glycyrrihizin on the oral epithelial cells remain unclear. Oral epithelial cells treated with glycyrrihizin in vitro or in vivo must be investigated further to elucidate this issue. Both 6-gingerol and 6-shogaol were introduced as natural products as the methylglyoxal-trapping materials, and they inhibited the generation of methylglyoxal-modified proteins and free AGEs, which were derived from methylglyoxal [40,41,175]. To inhibit the generation of intracellular AGEs in the oral epithelial cells, natural products need to directly show their function because target cells are exposed [10]. If natural products inhibit or suppress the generation or accumulation of intracellular AGEs in vivo, then their metabolism in the small/large intestine may have the same effect. However, we believe that natural products which inhibit the function or generation of intracellular AGEs in vitro, such as baicalin [169], liquiritigenin [36,170], liquiritin [36], glycyrrhizin [36,37], berberine [171], 6-gengerol [40,41], and 6-shogol [40,41,175], should be investigated further as treatment and prevention strategies for stomatitis because they can directly affect for the oral epithelial cells.

## 8. Natural Products Obtained from Hangeshashinto Water Extract Suppress Extracellular AGEs-RAGE/TLR4 Signaling in Stomatitis

Baicalin may suppress the AGE-RAGE signaling pathway in vitro and in vivo (Figure 3) [38,39]. Qui et al. reported that baicalin suppressed AGE-RAGE signaling in a DM animal model treated with streptozotocin [38]. In contrast, Fu et al. isolated and cultured porcine aortic vascular endothelial (PAVE) cells from ten 30-day-old naturally farrowed, caryly weaned piglets [39]. Baicalin suppresses RAGE signaling to modulate apoptosis, and blocking RAGE signaling may be effective when extracellular AGEs are agonists. Liquiritin attenuates AGEs-RAGE/NF-κβ in human umbilical vein endothelial cells in vitro (Figure 3) [176]. In contrast, liquiritin apioside and wogonin regulate the AGE-RAGE signaling pathway in diabetic mice (Figure 3) [177,178]. They were metabolized in various organs and bacteria following oral administration. However, the results indicate that liquiritin, apioside, and worgonin obtained from Hangeshashinto water extract may suppress AGE-RAGE signaling in oral epithelial cells. Isoliquiritigenin suppresses the AGE-RAGE signaling pathway in vitro and in vivo (Figure 4) [179,180]. Isoliquiritigenin was previously shown to ameliorate the toxicity of extracellular AGEs on cultured human renal proximal tubular epithelial cells [179]. This function of isoliquiritigenin may translate to oral epithelial cells, providing a preventive strategy for stomatitis. Because isoliquiririgenin was administered orally for db/db mice in the latter investigation, the resulting metabolites may inhibit the AGE-RAGE axis [180]. Shi et al. did not indicate if isoliquiritigenin or its metabolites exhibit this function; however, their investigation introduced the potential that isoliquiritigenin may suppress the AGE-RAGE axis in stomatitis. Goto et al. reported that berberine improves the high-mobility group box-1 (HMGB1)-RAGE or -TLR4 axis in a rat cardiomyocyte cell line (H9C2) in vitro (Figure 5) [181]. HMGB1 and other proteins can activate RAGE and TLR4 to induce inflammation and cell dysfunction, and AGE-modified proteins are contained within their agonists. In contrast, Jiang et al. reported that berberine modulated AGE-induced ferroptosis in both human keratinocyte cell lines (HaCaT cells) and the skin of db/db mice [182]. We predict that berberine will modulate AGE-induced ferroptosis in oral epithelial cells and their metabolites, which may exhibit similar effects. Fan et al. reported that glycyrrhizin inhibits the HMGB1-RAGE axis in various tumor cells (Figure 6) [183]. While Fan et al. did not suggest that glycyrrihizin blocked the AGEs-RAGE axis, it may show this function because RAGE can combine various types of AGE-modified proteins. Furthermore, 6-shogaol inhibited AGE-induced IL-6 and intracellular adhesion molecule 1 (ICAM1) expression on human gingival fibroblasts in vitro. The 6-shogaol suppressed AGE-induced RAGE expression and activation of MAPKs/NF-κβ signaling (Figure 6) [42]. We believe that 6-shogaol inhibits AGE-RAGE signaling, but the effect may not be direct, because the ratio of the AGE-RAGE combination may be reduced.

## 9. Beneficial Natural Products in Seven Crude Drugs Obtained from Hangeshashinto

Plant leaves, blanch, roots, seeds, and seed vessels contain high levels of natural products [184,185]. The various solvents (e.g., water, water/methanol, ethanol, acetone) that can be extracted from plants also contain natural products; however, they have not yet been fully elucidated. Oh et al. and Endo et al. identified approximately 20 components in Hangeshashinto water extract using 3D-HPLC [6,20]. The identified compounds are not necessarily present in high concentrations in the extract. Because researchers generally divide individual natural products using the retention times and absorption/excitation wavelengths with 3D-HPLC, the compounds whose retention times and wavelengths are different from those of other compounds are more likely to be isolated. The natural products in each crude drug (tuber of *Pinellia ternate* Breitenbach, Araceae; root of *Scutellaria baicalensis* Georgi, Labiatae; rhizome of *Coptis japonica* Makino, Ranueculaceae; steamed rhizome of *Zingiber officinale* Roscoe, Zingiberaceae; root of *Panax ginseng* C.A. Meyer, Araliaceae; the root or stolon of *Glycyrrhiza uralensis* Fischer, Laguminosae; and fruit of *Zizpbus jujura* Miller var. inermis Pehder, Rhamnaceae) obtained from the Hangeshashinto water extract must be further investigated (Table 1) [186]. We introduce natural products obtained from each extract of crude drug in Hangeshashinto, which were extracted with various solvent such as water, ethanol, and methanol, and review their functions, including the inhibition of the generation of intracellular AGEs, and suppression of extracellular AGEs-RAGE/TLR4 signaling (see Section 10 and Section 11), using the “Standards of Reporting Kampo Products (STORK)” database which was established by Japanese researchers [186].

## 10. Natural Products in the Crude Drugs Obtained from Hangeshashinto That Inhibit the Generation of Intracellular AGEs

Natural products in plants can suppress the generation of intracellular AGEs. We introduced quercetin, chrysin, genistein, (+)-catechin, (−)-epicatechin, epigarocatechin-3-gallate, hesperidin, imperialine, curcumin, piperine, diphylorethohydroxyarmalol, and resveratrol in Figure 14 [28,121,160,161,187]. These natural products are involved in the carbonyl trap system, where they remove AGE-origin compounds [28,121,160,161,187]. In contrast, curcumin and resveratrol inhibit glyoxalase-1 to promote the generation of glyoxal- and methylglyoxal-derived AGEs. However, the carbonyl trap system can trap them to suppress the generation of AGEs [162]. Although the main effects of curcumin and resveratrol for oral epithelial cells remain unclear, we believe that they should be considered as candidates for natural products for the suppression of AGE-induced cytotoxicity. We reviewed whether these natural products were detected and isolated in each crude drug obtained from Hangeshathinto (Table 2) [188,189,190,191,192,193,194,195,196,197,198,199,200,201,202,203,204,205,206].

## 11. Natural Products in Crude Drugs from Hangeshashinto Suppress Extracellular AGEs-RAGE/TLR4 Signaling

We introduce ferulic acid [214], caffeic acid [215], gallica acid [216,217], luteolin [218], apigenin [219,220], fisetin [221], naringenin [222], and naringin (naringenin-7-*O*-rhamunoglucoside) [223], which are natural products that have been obtained from various plants and can suppress/attenuate extracellular AGEs-RAGE/TLR4 signaling and cytotoxicity (Figure 15). Although quercetin and *p*-coumaric acid were introduced as natural products that inhibited the generation of intracellular AGEs in Section 10 (Figure 14a,g), both quercetin [224,225,226,227,228,229,230] and *p*-coumaric acid [231,232,233,234] were able to suppress extracellular AGE-induced cytotoxicity via RAGE/TLR4. We investigated the natural products in each crude drug obtained from Hangeshashinto and introduced this information in Table 3 [235,236,237,238,239,240,241,242,243,244,245,246]. Naringenin is an aglycone type, and naringin is a glycosylation type of naringenin [223]. Although naringenin and naringin have been isolated from various citruses [247,248], we cannot observe a reference that naringin was isolated in seven crude drugs in Hangeshashinto. However, naringin-4‘-*O*-glucoside was isolated from *Glycyriza glabra*, and it is expected that naringin also will be isolated due to their structural similarities [249].

## 12. Limitations

While it is likely that AGE-induced oral squamous cell syndromes occur as a result of modern lifestyles, we targeted the intracellular AGEs in the oral epithelial cells and extracellular AGEs that directly combine with the surface. AGEs in other organs can promote cytotoxicity for oral epithelial cells via inflammation and saliva dysfunction; however, we were unable to review this phenomenon. Various natural products are generally contained in the roots, leaves, radix, and seeds of the plants [10,184,185]. Although the analysis of the Hangeshashinto water extract shows that it contains high amounts of natural products, we have only discussed nineteen compounds, and we did not present the ratio of these compounds in the water extract weight (e.g., μg/g dry weight). Therefore, we cannot accurately assess their bioactivity for anti-intra-/extracellular cytotoxicity in oral epithelial cells. Moreover, we did not review whether whole natural products show anti-intracellular and extracellular AGE-induced cytotoxicity in oral epithelial cells because these issues will be addressed as detection and isolation technologies develop for natural products in the Hangeshashinto extract. In contrast, we reviewed some natural products that show anti-AGE-induced cytotoxicity in seven crude drugs in Hangeshashinto. These natural products are typical compounds for anti-AGE materials in various plants. If we introduce the major compounds in each crude drug, then Section 9 and Section 10 would contain details for over 200 compounds [186], which is beyond the scope of this review. Therefore, we introduced some natural products that show anti-AGE-induced cytotoxicity and are contained in various plants, and reviewed whether they are included in each crude drug in Section 9 and Section 10. This information should suggest the possibility that more natural products, which suppress the AGE-induced cytotoxicity in oral epithelial cells, will be isolated in the Hangeshashinto water extract.

## 13. Conclusions

While intra-/extracellular AGEs may induce oral squamous cell syndromes, these syndromes have not been recognized in clinical or basic medical science. Hangeshashinto treatments are generally administered orally, and the mechanisms by which they attenuate the mouth, esophagus, and gut have been widely investigated. This review focused on treatments of oral epithelial cells that involve gargling and coating the mouth with Hangeshashinto water extract. Nineteen natural products that can suppress the generation of intracellular AGEs and extracellular AGEs-RAGE/TLR4 signaling have been isolated from the Hangeshashinto water extract. In addition, sixteen natural compounds in seven crude drugs obtained from Hangeshashinto may show anti-intra-/extracellular AGE effects. To inhibit the generation of various types of AGE-modified proteins, the carbonyl trap system and activation of glyoxalase-1, which are natural products in the water extract of Hangeshashinto or seven crude drugs, are beneficial because they can react with AGE-origin compounds. Although they can suppress the signaling of AGEs in the body fluid or the beverages/foods and RAGE/TLR4 in oral epithelial cells, the possibility that they can suppress various or whole types of AGEs-RAGE/TLR4 signaling remains unclear. Each natural compound may suppress an individual structure in the AGE-RAGE/TLR4 axis; this will require further investigation. Natural products in Hangeshashinto water extract may directly prevent and modulate stomatitis induced by intra-/extracellular AGEs. This review will be beneficial for future investigations that aim to reveal the mechanisms by which the Hangeshashinto water extract attenuates AGE-induced stomatitis.

## Figures and Tables

**Figure 1 ijms-26-09118-f001:**
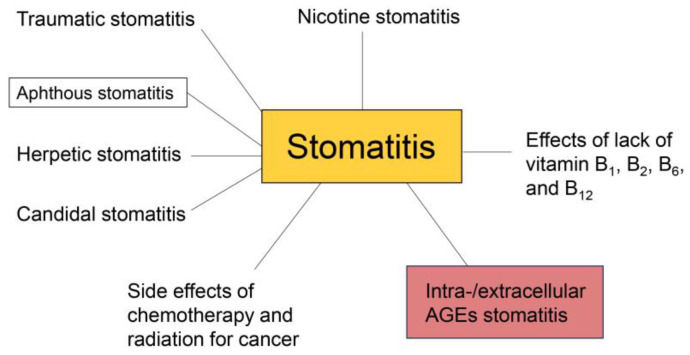
Types of stomatitis. Traumatic stomatitis [1,48], aphthous stomatitis [2,49], herpetic stomatitis [50,51], candidal stomatitis [52,53], nicotine stomatitis [54,55], lack of vitamin B_1_, B_2_, B_6_, and B_12_ [56,57], and intra-/extracellular AGEs stomatitis are described in this review (see Section 6). The black line indicates that each case of stomatitis is confined to the clinical stage.

**Figure 2 ijms-26-09118-f002:**
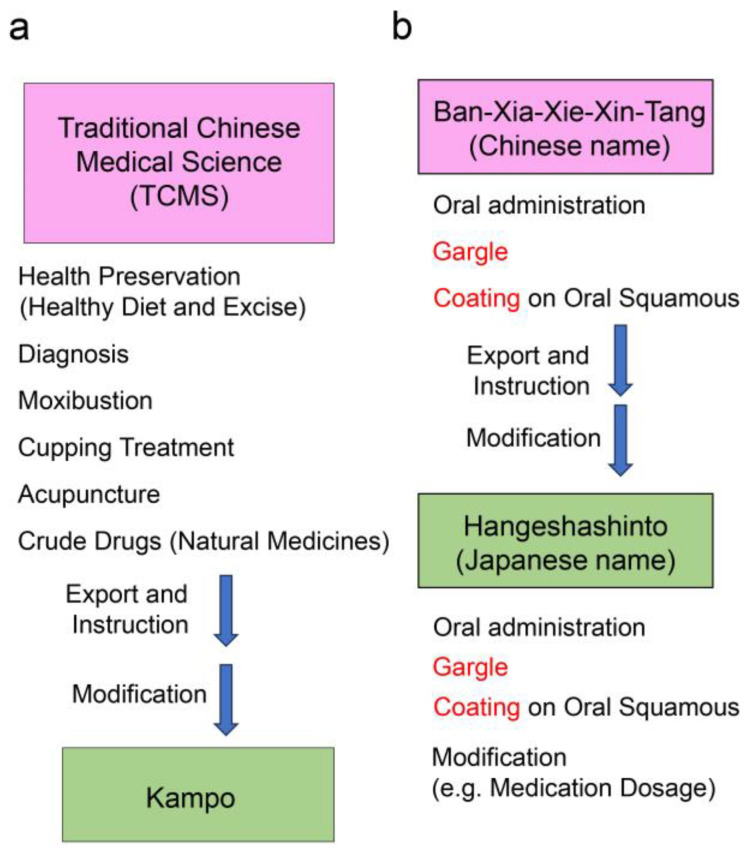
Relationship between traditional Chinese medical science (TCMS) and Kampo, and specifically, the Hangeshashinto (Ban-Xia-Xie-Xin-Tang) treatment [5,6,7,8,11,59,60,61,62]. (**a**) TCMS was exported and instructed into antient Japan from antient China, and was modified to “Kampo” in Japan. (**b**) Ban-Xia-Xie-Xin-Tang and the infrmomation of the method of treatment were exported and instructed into antient Japan from antient China. Ban-Xia-Xie-Xin-Tang (Chinese name) had been called Hangeshashinto in as Japanese name. Pink and green squares indicate TCMS and Kampo (traditional Japanese medicine), respectively. Ban-Xia-Xie-Xin-Tang and Hangeshashinto are composed of the same components. Hangeshashinto is generally administered as an oral treatment; however, gargling and coating of the oral squamous cells were used in ancient practice. Blue arrow indicate the medical science, medicine, and information of treatment of antient China were exported and instructed into antient Japan, and they were modified in Japan. Red term indicate the important and focued treatment compared with other treatment of the traditional Chinese and Japanese meidicines (See Section 3.2).

**Figure 3 ijms-26-09118-f003:**
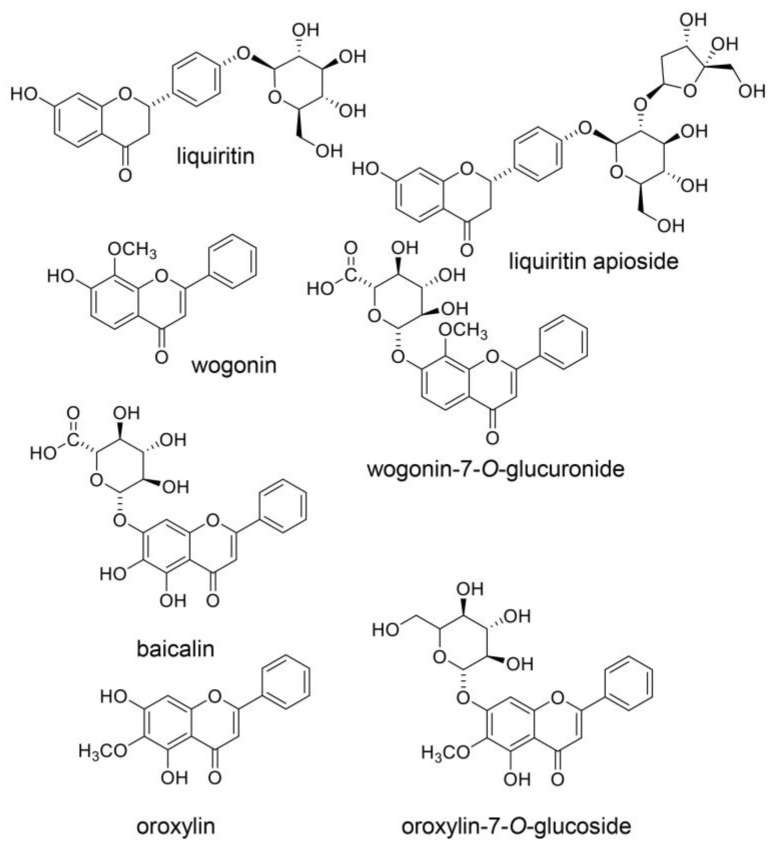
Flavonoid compounds obtained from the Hangeshathito water extract [6,20]. Liquiritin [6], liquiritin apioside [6], wogonin [6,20], worgonin-7-*O*-glucuronide [6], baicalin (baicalein-7-*O*-glucuronide) [6], oroxylin [6], and oroxylin-7-*O*-glucoside [6].

**Figure 4 ijms-26-09118-f004:**
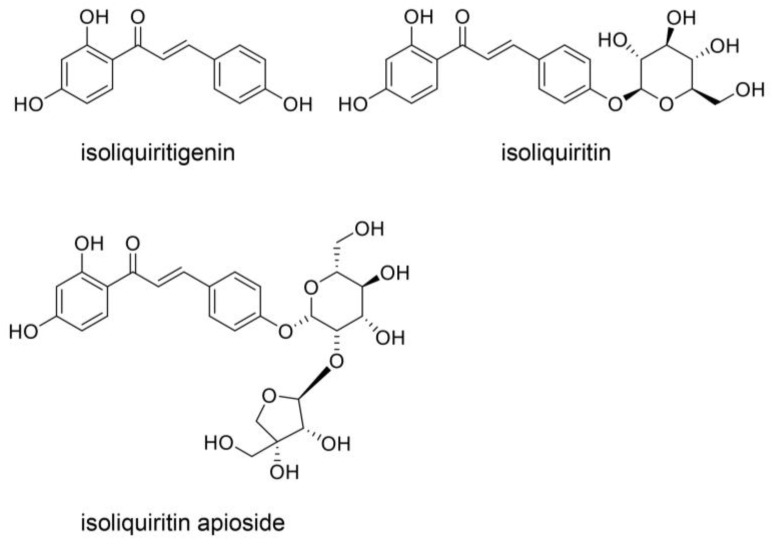
Chalcon compounds identified in the Hangeshashinto water extract [6]. Isoquritin [6], isoliquiritin apioside [6], and isoquitigenin [6].

**Figure 5 ijms-26-09118-f005:**
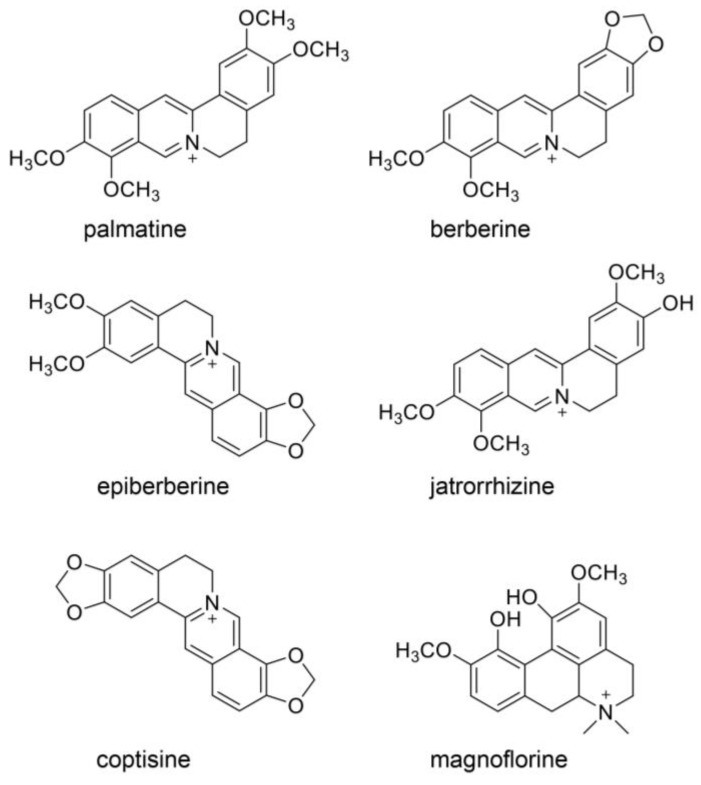
Alkaloid compounds obtained from the Hangeshathito water extract [6]. Berberine [6], epiberberine [6], jateprrhizine [6], coptisine [6], and magnoflorine [6].

**Figure 6 ijms-26-09118-f006:**
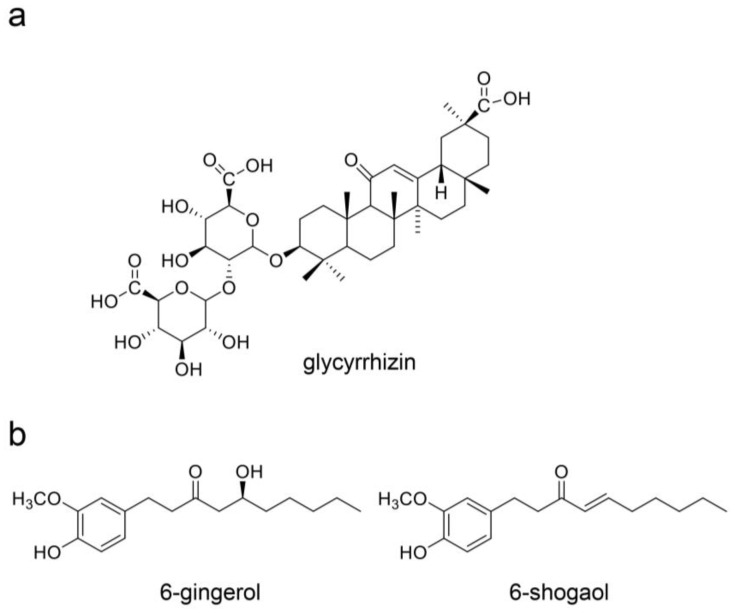
Triterpene and gingerol groups (one of the monophenolic acid groups) obtained from the Hangeshashinto water extract. [6,20,22]. (**a**) Glycyrrhizin [6,22], (**b**) 6-gingerol [20], and 6-shogaol [6].

**Figure 7 ijms-26-09118-f007:**
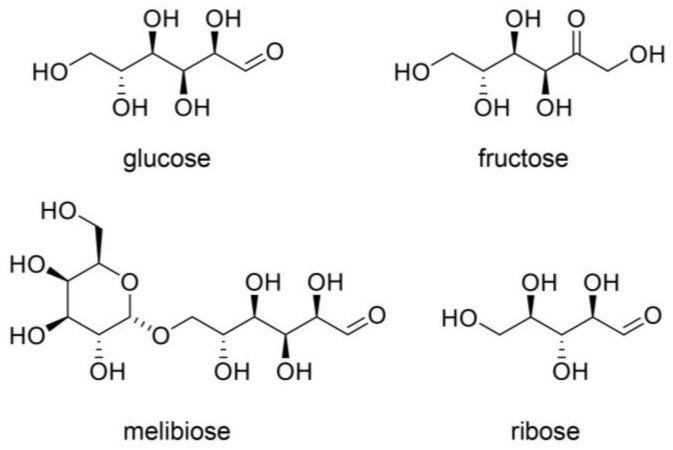
Major saccharides involved in the generation of AGEs. Glucose [26,27,28,29,93,94], fructose [26,27,28,29,93,94], melibiose [95], and ribose [96,97].

**Figure 8 ijms-26-09118-f008:**
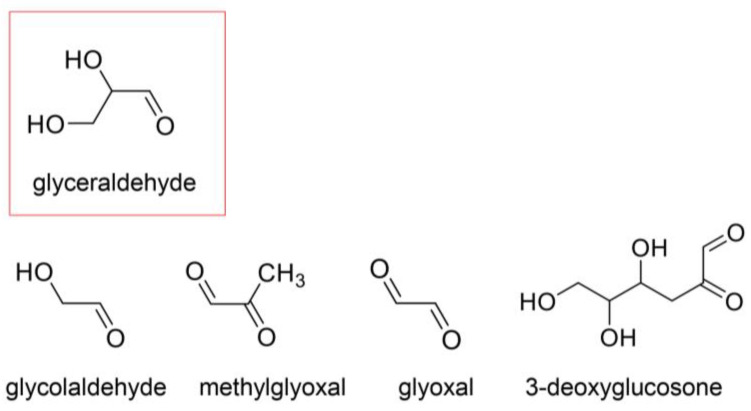
The metabolites and non-enzymatic reaction products of glucose, fructose, and ribose. AGEs are generated from glyceraldehyde, glycolaldehyde, methylglyoxal, glyoxal, and 3-deoxyglucosone [26,27,28,29]. The open red square identifies a triose.

**Figure 9 ijms-26-09118-f009:**
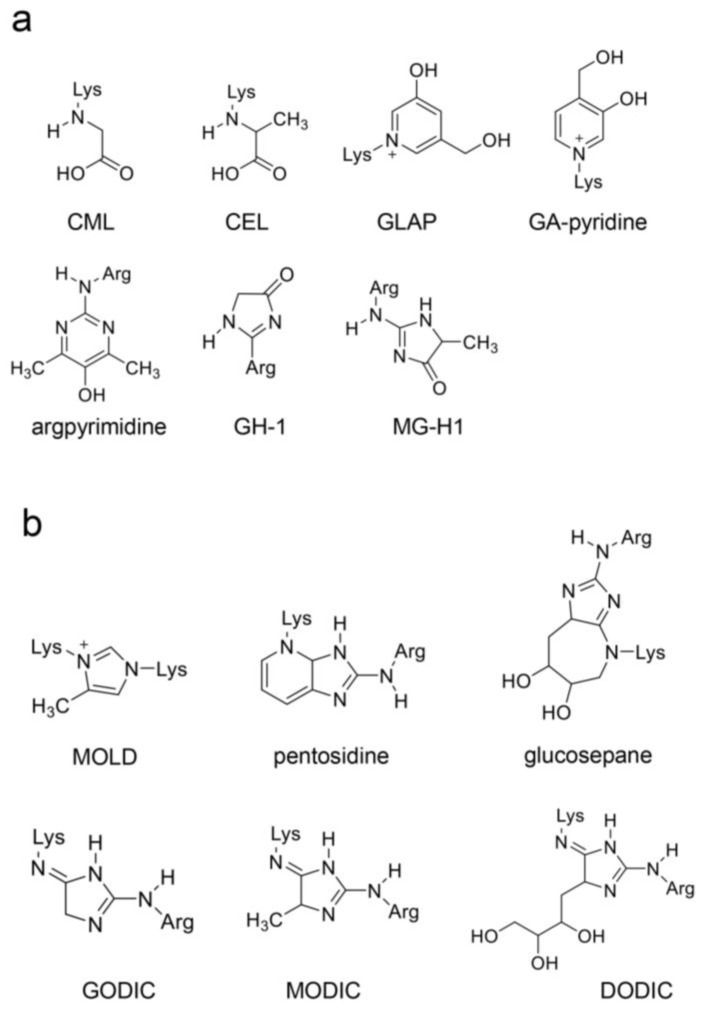
Free-type AGEs [26,27,28,29]. (**a**) Free-type AGEs with one amino acid residue. CML, *N*^ε^-carboxymethyl–lysine; CEL, *N*^ε^-carboxyethyl–lysine; GLAP, glyceraldehyde-related pyridinium; GA-pyridine; argpyrimidine; GH-1, *N*^δ^-(5-hydro-4-imidazolone-2-yl)-ornithine (glyoxal-hydro-imidazolone); MG-H1, *N*^δ^-(5-hydro-5-methyl-4-imidazolone-2-yl)-ornithine (methylglyoxal-hydro-imidazolone). (**b**) Free-type AGEs with two amino acid residues. MOLD, methylglyoxal-derived imidazolium cross-link (methylglyoxal–lysine dimer); pentosidine; glucospane; GODIC, glyoxal-derived imidazolium cross-link; MODIC, methylglyoxal-derived imidazolium cross-link; and DODIC, 3-doxyglucosone-derived imidazolium cross-link.

**Figure 10 ijms-26-09118-f010:**
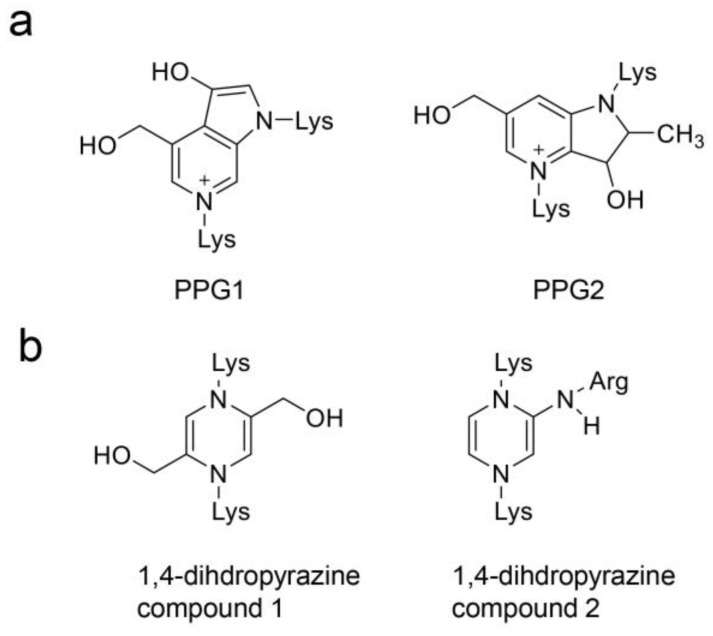
Free-type AGEs were synthesized in a tube, and their structure was predicted [104,105]. (**a**) PPG1, pyrrolopyridinium–lysine dimer-derived glyceraldehyde 1; PPG2, pyrrolopyridinium–lysine dimer-derived glyceraldehyde 2. They were synthesized from glyceraldehyde and *N*^α^-acetyl-lysine in the tube in 2020, but were not detected in vivo in 2025 [104]. (**b**) 1,4-dihydropyrazine compounds 1 and 2 were named as “Toxic AGEs (TAGE)” by Takeuchi et al. They hypothesized the structure of TAGE; however, it has not been proved [105].

**Figure 11 ijms-26-09118-f011:**
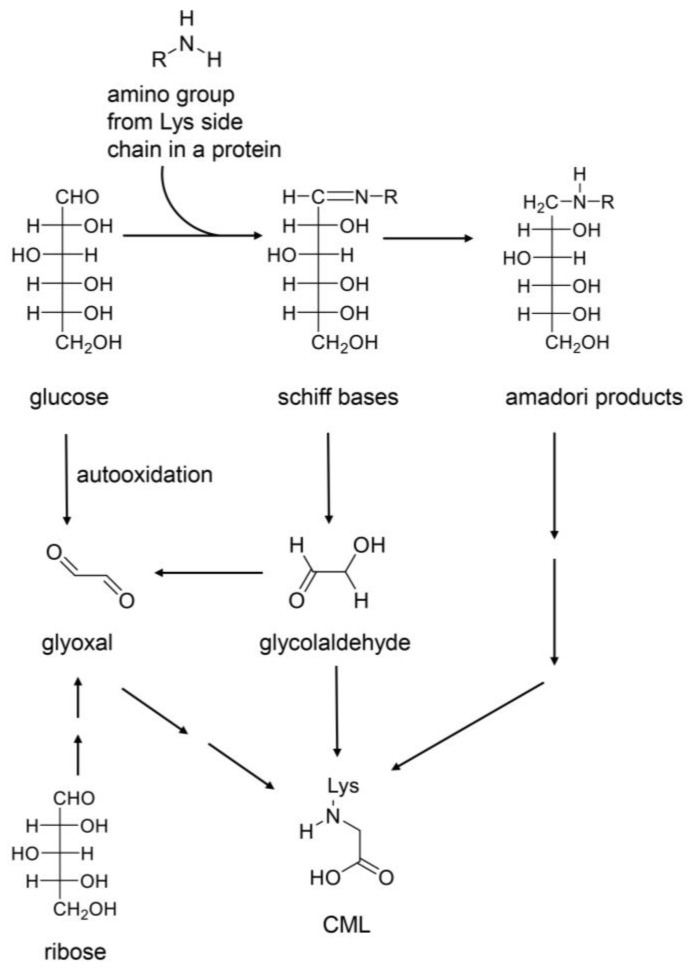
Various routes in the production of CML from glucose and ribose [28,96,97]. Glyoxal and glycolaldehyde are produced through enzymatic reaction and autooxidation. CML, *N*^ε^-carboxymethyl–lysine.

**Figure 12 ijms-26-09118-f012:**
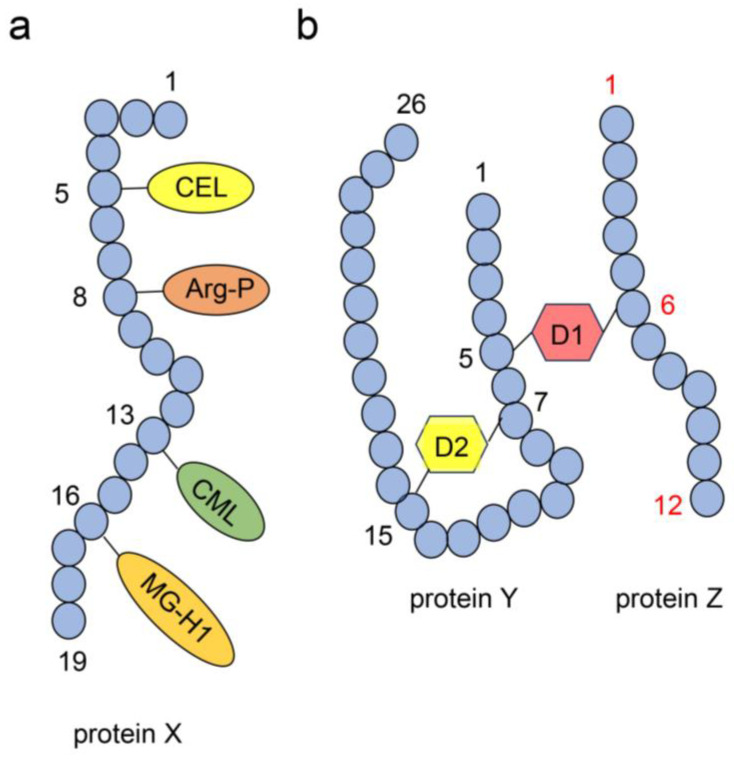
AGE-modified proteins [27,28,29]. Blue circles indicate the amino acids in proteins X, Y, and Z. Black and red indicate the number of amino acids. (**a**) AGEs, including free types, can be modified into one molecular protein, X. CEL, Nε-cassrboxyethyl-lysine; Arg-P, argpyrimidine; CML, Nε-carboxymethyl–lysine; MG-H1, *N*^δ^-(5-hydro-5-methyl-4-imidazolone-2-yl)-ornithine (methylglyoxal-hydro-imidazolone). (**b**) D1 and 2; model of free-type AGEs that contains two amino acid residues. Proteins Y and Z are linked with D1 via an intermolecular covalent bond. D1 connected both 5th in protein Y and 6th in protein Z. In contrast, the 7th and 15th amino acids in protein Y are associated with D2 via an intramolecular covalent bond.

**Figure 13 ijms-26-09118-f013:**
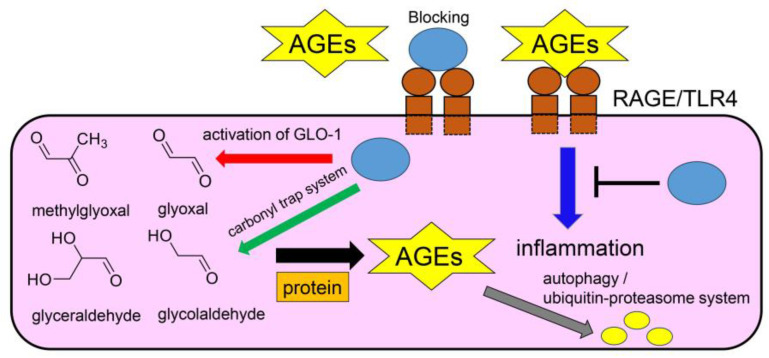
Mechanisms of inhibition of intra-/extracellular AGE-induced cytotoxicity [27,28,29]. The pink square represents a cell. Blue circles indicate anti-AGE compounds. Brown circles and squares indicate full-length RAGE or TLR4. Orange squares indicate protein. The yellow hexagrams indicate AGEs. The yellow circles indicate degraded AGEs. Black arrows indicate the Maillard reaction and other non-enzymatic reactions involving proteins and AGE precursors (e.g., methylglyoxal, glyoxal, glyceraldehyde, glycolaldehyde). The red arrow indicates activation of GLO-1. The green arrow indicates the carbonyl trap system. The gray arrow indicates AGE degradation via autophagy or the ubiquitin–proteasome system. The blue arrow indicates a cell signaling pathway to induce inflammation. The black line indicates inhibition effects. AGE, advanced glycation end-products; RAGE, receptor for advanced glycation end-products; TLR4, toll-like receptor 4. GLO-1, glyoxalase-1.

**Figure 14 ijms-26-09118-f014:**
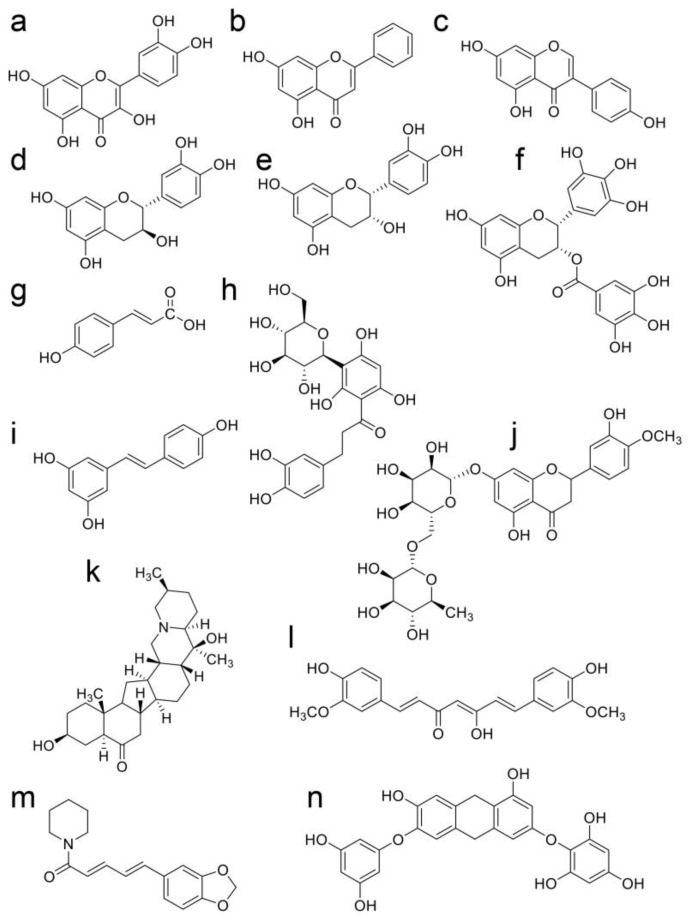
Natural products obtained from various plants that inhibit the generation of intracellular AGEs [28,121,160,161,187]: (**a**) quercetin, (**b**) chrisin, (**c**) genistein, (**d**) (+)-catechin, (**e**) (−)-epicatechin, (**f**) epigallocatechin-3-gallate, (**g**) *p*-hydroxy cinnamic acid (*p*-coumaric acid), (**h**) aspalatin, (**i**) resveratrol, (**j**) hesperidin, (**k**) imperialine, (**l**) curcumin, (**m**) piperine, and (**n**) diphlorethohydroxycarnalol.

**Figure 15 ijms-26-09118-f015:**
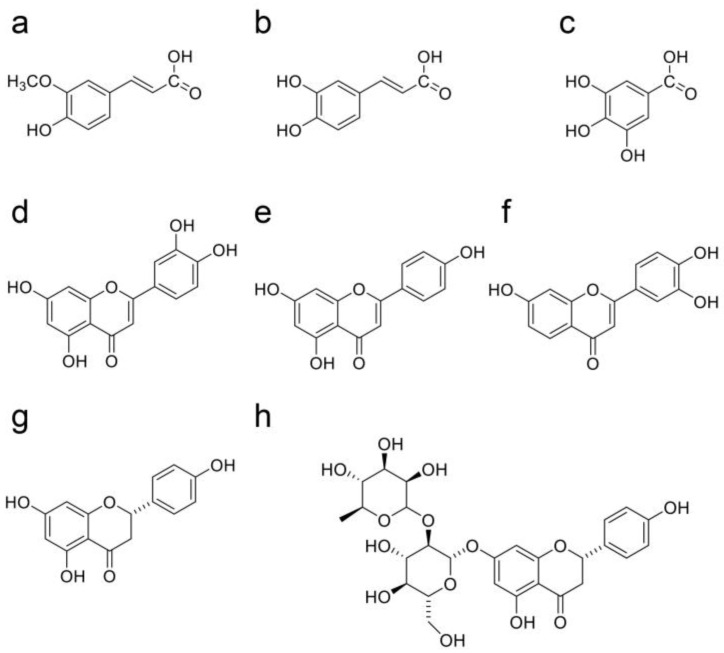
Natural products in various plants that inhibit extracellular AGEs-RAGE/TLR4 signaling to attenuate cytotoxicity [28,121,160,161,187]; (**a**) ferulic acid, (**b**) caffeic acid, (**c**) gallic acid, (**d**) luteolin, (**e**) apigenin, (**f**) fisetin, (**g**) naringenin, and (**h**) naringin.

**Table 1 ijms-26-09118-t001:** Crude drugs obtained from Hangeshashinto, including Hange, Ogon, Oren, Kankyo, Ninjin, Kanzo, and Taiso. The names of crude drugs are described using Japanese, English, and Latin names.

No.	Japanese Name	English Description	Description of Latin Name
1	Hange	Pinellia tuber	Tuber of *Pinellia ternate* Breitenbach, Araceae
2	Ogon	Scutellaria root	Root of *Scutellaria baicalensis* Georgi, Labiatae
3	Oren	Coptidis rhizome	Rhizome of *Coptis japonica* Makino, Ranueculaceae
4	Kankyo	Ginger rhizome	Steamed rhizome of *Zingiber officinale* Roscoe, Zingiberaceae
5	Ninjin	Ginseng root	Root of *Panax ginseng* C.A. Meyer, Araliaceae
6	Kanzo	Glycyrrhiza root	Root or stolon of *Glycyrrhiza uralensis* Fischer, Laguminosae
7	Taiso	Jujube fruit	Fruit of *Zizpbus jujura* Miller var. inermis Pehder, Rhamnaceae

**Table 2 ijms-26-09118-t002:** Crude drugs obtained from Hangeshashinto and their natural products can inhibit the generation of the intracellular AGEs to attenuate cytotoxicity. The crude drug names are described using Japanese and Latin names.

No.	Japanese Name	Description of Latin Name	Natural Product	Reference
1	Hange	Tuber of *Pinellia ternate* Breitenbach, Araceae	No information	No information
2	Ogon	Root of *Scutellaria baicalensis* Georgi, Labiatae	quercetin	[188,189]
			genistein	[190,191]
			(+)-catechin	[192,193]
			epigallocatechin-3-gallate	[194]
			hesperidin	[195]
			*p*-coumaric acid	[196]
			curcumin	[195,197]
3	Oren	Rhizome of *Coptis japonica* Makino, Ranueculaceae	quercetin	[198]
4	Kankyo	Steamed rhizome of *Zingiber officinale* Roscoe, Zingiberaceae	No information	No information
5	Ninjin	Root of *Panax ginseng* C.A. Meyer, Araliaceae	quercetin	[199,200]
			(+)-catechin	[201,202]
			epigallocatechin-3-gallate	[203,204]
			*p*-coumaric acid	[205,206]
6	Kanzo	Root or stolon of *Glycyrrhiza uralensis* Fischer, Laguminosae	quercetin	[43,207]
			genistein	[45]
			(+)-catechin	[208]
			epigallocatechin-3-gallate	[44,209]
			*p*-coumaric acid	[210]
7	Taiso	Fruit of *Zizpbus jujura* Miller var. inermis Pehder, Rhamnaceae	quercetin	[211]
			(+)-catechin	[212]
			(−)-epicatechin	[212]
			*p*-coumaric acid	[213]

**Table 3 ijms-26-09118-t003:** Crude drugs in Hangeshashinto and their natural products that can inhibit extracellular AGEs-RAGE/TLR4 signaling to attenuate cytotoxicity. The names of crude drugs are described using Japanese and Latin names.

No.	Japanese Name	Description of Latin Name	Natural Product	Reference
1	Hange	Tuber of *Pinellia ternate* Breitenbach, Araceae	No information	No information
2	Ogon	Root of *Scutellaria baicalensis* Georgi, Labiatae	fisetin	[235]
			luteolin	[236]
			quercetin	[188,189]
			*p*-coumaric acid	[196]
3	Oren	Rhizome of *Coptis japonica* Makino, Ranueculaceae	luteolin	[237]
			quercetin	[198]
4	Kankyo	Steamed rhizome of *Zingiber officinale* Roscoe, Zingiberaceae	No information	No information
5	Ninjin	Root of *Panax ginseng* C.A. Meyer, Araliaceae	ferulic acid	[238]
			caffeic acid	[205,239]
			gallic acid	[47,240]
			naringenin	[241]
			quercetin	[199,200]
			*p*-coumaric acid	[205,208]
6	Kanzo	Root or stolon of *Glycyrrhiza uralensis* Fischer, Laguminosae	caffeic acid	[242,243]
			gallic acid	[244]
			apigenin	[245]
			naringenin	[246]
			quercetin	[43,207]
			*p*-coumaric acid	[210]
7	Taiso	Fruit of *Zizpbus jujura* Miller var. inermis Pehder, Rhamnaceae	quercetin	[211]
			*p*-coumaric acid	[212]

## Data Availability

The data presented in this study are available in the article.

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
