# Peer review of "Potential of Natural Products in Hangeshashinto Water Extract on the Direct Suppression of Stomatitis Induced by Intra-/Extracellular Advanced Glycation End-Products"

_ijms, 2025, doi:10.3390/ijms26189118_

Round 1

Reviewer 1 Report

Comments and Suggestions for Authors

The manuscript covers an important aspect of use of natural products in management of Advance glycation end products (AGEs). I have noticed few issue that needs addressal

  1. Title is grammatically in correct. Plz check
  2. In abstract and at numerous place in manuscript, there is un necessary use of commas ”,”. Please check carefully and use only where needed.
  3. Abstract line 48-49 “This review summarizes the natural products identified in the Hangeshashinto water extract and crude drugs”  this sentence needs clarification
  4. Line 58-60 “There are various types of stomatitis, such as traumatic and aphthous stomatitis, stomatitis as a side effect of chemotherapy/radiotherapy [1–4], and stomatitis that results 59 from a lifestyle-related disease (LSRD).” This senetence is not Correct grammatically. The manuscript is full of such errors and needs a careful check by native English speaker.
  5. Heading 2 seems strange and it can be more clearly written as “Various types of Stomatits” or any other suitable heading. Please check all headings.
  6. Although the manuscript theme area is Advance glycation end products (AGEs), little or no detailed information is given in Introduction part about AGEs. Authors must add information about their generation in body, their major types and hazrads. I can see this in details but at least a short paragraph must be added in introduction too.
  7. Authors must give information regarding mechanism of stomatitis
  8. I would further suggest a graphical representation of Effects of Hangeshashinto compounds on AGEs.

Author Response

Response Letter to Reviewers’ Comments

Responses to Reviewer 1

Dear Reviewer 1:

Thank you for giving us the opportunity to submit a revised draft of our manuscript titled “Natural Products in Hangeshashinto (Ban-Xia-Xie-Xin-Tang) Water Extract May Prevent and Modulate Stomatitis Induced by Advanced Intra-and Extracellular Glycation End-Products” to the International Journal of Molecular Sciences (manuscript ID: ijms-3857167). We appreciate the time and effort you have taken to provide valuable feedback on our manuscript; your comments have enriched the manuscript and helped us to produce a more balanced account of our research. Please note that the manuscript has also been reviewed by a professional English language editor (Editage) to address all grammatical and syntax errors and improve the overall readability of the document.

The following minor revisions were made to the manuscript:

  • The article title has been changed (Previous title: Potential of Natural Products in Hangeshasnito (Ban-Xia-Xie-Xin-Tang) Water Extract on the Direct Suppressing of Stomatitis Induced by Intra-/Extracellular Advanced Glycation End-Products).
  • Figures 1,2, 4, and 11 were revised, and Figure 13 was added.
  • A new section, Section 5, has been added to explain the mechanisms by which various anti-AGE compounds that contain natural products inhibit cytotoxicity via intra/extracellular AGEs. The new section includes the newly added Figure 13.

Comments and Suggestions for Authors

The manuscript covers an important aspect of use of natural products in management of Advance glycation end products (AGEs). I have noticed few issue that needs addressal

Main points:

Comment 1 : Title is grammatically in correct. Plz check.

Response 1: We have revised the title accordingly.

Comment 2 : In abstract and at numerous place in manuscript, there is un necessary use of commas”,”. Please check carefully and use only where needed.

Response 2: The manuscript has been carefully revised by ourselves and a professional English language editor (Editage) to address all issues with grammar and readability.

Comment 3 : Abstract line 48-49 “This review summarizes the natural products identified in the Hangeshashinto water extract and crude drugs” This sentence needs clarification.

Response 3: We have revised the abstract accordingly and included the following sentence:

“This review summarizes 19 natural products identified in the Hangeshashinto water extract and 16 natural products identified in the crude drug extract.”

We have also modified sentences in the Conclusion section to ensure that we describe that nineteen natural compounds in the Hangeshashinto water extract and sixteen natural compounds in the seven crude drugs that were identified in Hangeshashinto and may have anti-AGE effects.

Comment 4: Line 58-60 “There are various types of stomatitis, such as traumatic and aphthous stomatitis, stomatitis as a side effect of chemotherapy/radiotherapy [1–4], and stomatitis that results 59 from a lifestyle-related disease (LSRD).” This senetence is not Correct grammatically. The manuscript is full of such errors and needs a careful check by native English speaker.

Response 4: We have revised these sentences and ensured that they are reviewed by a professional English language editor (Editage).

Comment 5 : Heading 2 seems strange and it can be more clearly written as “Various types of Stomatits” or any other suitable heading. Please check all headings.

Response 5: We have modified Heading 2 based on the reviewer’s comment. The New Heading 2 is “Various types of Stomatitis”

Comment 6 : Although the manuscript theme area is Advance glycation end products (AGEs), little or no detailed information is given in Introduction part about AGEs. Authors must add information about their generation in body, their major types and hazrads. I can see this in details but at least a short paragraph must be added in introduction too.

Response 6: We have described AGE origins, their generation in the human body, major types, and cytotoxicity in the revised Introduction.

Comment 7 : Authors must give information regarding mechanism of stomatitis.

Response 7: We have described the information regarding the mechanisms of stomatitis in Section 2 of the revised manuscript.

Comment 8 : I would further suggest a graphical representation of Effects of Hangeshashinto compounds on AGEs.

Response 8: Based on the reviewer’s comment, we inserted a new section and figure, Section 5 and Figure 13, in the revised manuscript. In Section 5 we describe general anti-AGE compounds that contain natural products that suppress cytotoxicity by intra-/extracellular AGEs. We also introduce information regarding the components in Hangeshashinto water extract that may inhibit the generation and accumulation of intracellular AGEs and suppress AGEs-RAGE/TLR4 signaling, which induces cytotoxicity, such as inflammation, in Sections 6 and 7.

Reviewer 2 Report

Comments and Suggestions for Authors

This manuscript reviews various monomeric molecules isolated and identified from the aqueous extract of Ban-Xia-Xie-Xin-Tang and explores their therapeutic potential for stomatitis, particularly that induced by advanced glycation end products (AGEs). In recent years, several review articles have summarized the monomeric natural products derived from Ban-Xia-Xie-Xin-Tang and their biological activities. However, little attention has been paid to the treatment of stomatitis by Ban-Xia-Xie-Xin-Tang through the inhibition of AGEs formation and interference with cell signal transduction. Therefore, this manuscript offers a valuable reference for future research in this area.

If the authors could provide a review of all the monomer molecules reported in the water extract of Ban-Xia-Xie-Xin-Tang, it would greatly enhance the reference value of this manuscript. However, the authors have already pointed out the difficulty of this issue in "11. Limitations", so I would like to raise the following issues that need improvement:

1. Figures 1 and 2, especially Figure 2, need improvement. The expression in Figure 2 is a bit chaotic. Logically, the elements should be arranged from top to bottom. Besides, please align all parts. It seems better to place "Stomatitis" in the middle rather than at the top in Figure 1. In addition, it would be better if the picture were drawn in color.

2. Some formatting issues require a full-text review. For example, in Line 155, the extra "]" should be deleted.

3. The first letters of the compound names in the figures are sometimes capitalized and sometimes not. Please make them consistent.

Author Response

Response Letter to Reviewers’ Comments

Responses to Reviewer 2

Dear Reviewer 2:

Thank you for allowing us to submit a revised draft of our manuscript titled “Natural Products in Hangeshashinto (Ban-Xia-Xie-Xin-Tang) Water Extract May Prevent and Modulate Stomatitis Induced by Advanced Intra-and Extracellular Glycation End-Products” to the International Journal of Molecular Sciences (manuscript ID: ijms-3857167). We appreciate the time and effort you have taken to provide valuable feedback on our manuscript; your comments have enriched the manuscript and helped us produce a more balanced account of our research. The manuscript has been reviewed by a professional English language editor (Editage) to address all grammatical and syntax errors and improve the overall readability of the document.

The following minor revisions were made to the manuscript:

  • The article title has been changed (Previous title: Potential of Natural Products in Hangeshasnito (Ban-Xia-Xie-Xin-Tang) Water Extract on the Direct Suppressing of Stomatitis Induced by Intra-/Extracellular Advanced Glycation End-Products).
  • Figures 1,2, 4, and 11 were revised, and Figure 13 was added.
  • A new section, Section 5, has been added to explain the mechanisms by which various anti-AGE compounds that contain natural products inhibit cytotoxicity via intra/extracellular AGEs. The new section includes the newly added Figure 13.

Comments and Suggestions for Authors

This manuscript reviews various monomeric molecules isolated and identified from the aqueous extract of Ban-Xia-Xie-Xin-Tang and explores their therapeutic potential for stomatitis, particularly that induced by advanced glycation end products (AGEs). In recent years, several review articles have summarized the monomeric natural products derived from Ban-Xia-Xie-Xin-Tang and their biological activities. However, little attention has been paid to the treatment of stomatitis by Ban-Xia-Xie-Xin-Tang through the inhibition of AGEs formation and interference with cell signal transduction. Therefore, this manuscript offers a valuable reference for future research in this area.

If the authors could provide a review of all the monomer molecules reported in the water extract of Ban-Xia-Xie-Xin-Tang, it would greatly enhance the reference value of this manuscript. However, the authors have already pointed out the difficulty of this issue in "11. Limitations", so I would like to raise the following issues that need improvement:

Main points:

Comment 1 : Figures 1 and 2, especially Figure 2, need improvement. The expression in Figure 2 is a bit chaotic. Logically, the elements should be arranged from top to bottom. Besides, please align all parts. It seems better to place "Stomatitis" in the middle rather than at the top in Figure 1. In addition, it would be better if the picture were drawn in color.

Response 1: Based on the reviewer’s comments, we have revised Figures 1 and 2 and added some information to the legends.

Comment 2 : Some formatting issues require a full-text review. For example, in Line 155, the extra "]" should be deleted.

Response 2: We have checked and corrected the formatting issues, and the professional English language editor (Editage) has also reviewed these issues.

Comment 3 : The first letters of the compound names in the figures are sometimes capitalized and sometimes not. Please make them consistent.

Response 3: We have corrected the formatting throughout the figure legends.
